# Study on Flame Retardancy Behavior of Epoxy Resin with Phosphaphenanthrene Triazine Compound and Organic Zinc Complexes Based on Phosphonitrile

**DOI:** 10.3390/molecules28073069

**Published:** 2023-03-30

**Authors:** Bo Xu, Menglin Wu, Yanting Liu, Simiao Wei

**Affiliations:** 1School of Chemistry and Materials Engineering, Beijing Technology and Business University, No. 11, Fucheng Road, Haidian District, Beijing 100048, China; 2Beijing Key Laboratory of Quality Evaluation Technology for Hygiene and Safety of Plastics, No. 11, Fucheng Road, Haidian District, Beijing 100048, China; 3China Light Industry Engineering Technology Research Center of Advanced Flame Retardants, No. 11, Fucheng Road, Haidian District, Beijing 100048, China; 4Petroleum and Chemical Industry Engineering Laboratory of Non-Halogen Flame Retardants for Polymers, No. 11, Fucheng Road, Haidian District, Beijing 100048, China; 5China Metallurgical Information and Standardization Research Institute, No. 11, Dengshikou Street, Dongcheng District, Beijing 100730, China

**Keywords:** flame retardant, organozinc complexes, phosphaphenanthrene, epoxy resins

## Abstract

A novel flame retardant phosphorus-containing organozinc complex (Zn-PDH) was prepared using zinc and iron as the metal center and 4-aminopyridine, with low steric hindrance, as the organic ligand, then using phosphazene to modify the organometallic complex (Zn-4APD). The flame retardant properties and mechanism of Zn-PDH/Tris-(3-DOPO-1-propyl)-triazinetrione (TAD) in epoxy resin (EP) were investigated. Flame inhibition behavior was studied by the vertical combustion test (UL94), while limiting oxygen index (LOI) measurement and flame retardant properties were studied by the cone calorimeter test (CONE). The flame retardant modes of action were explored by using the thermogravimetry–Fourier transform infrared (TG-FTIR) test, X-ray photoelectron spectrometer (XPS), and Raman spectroscopy (LRS). When TAD and Zn-PDH were added to the epoxy resin in the ratio of 3:1, the system achieved a balance between the gas-phase and condense-phase actions of the flame retardant effects, and the 3%TAD/1%Zn-PDH/EP composite system achieved not only good flame inhibition but also obtained good smoke and heat suppression performance, showing a comprehensive flame retardant performance. The gas phase and Zn-PDH mostly promoted charring with a barrier and protective effect in the condensed phase. As for the mechanism, TAD released the phosphorus-containing radicals and phenoxy radicals during decomposition and mainly exerted a gas-phase quenching effect. While in the condense phase, Zn-PDH promoted the decomposition of the polymer matrix to produce more aromatic structures and rapidly formed a complete and dense carbon layer rich in P-O-C crosslinked structures at high temperatures. Meanwhile, more N entered the gas phase in the form of inert gas, which diluted the concentration of the combustible fuel and helped to inhibit the combustion reaction.

## 1. Introduction

Due to their good electrical properties, bond strength, mechanical properties, chemical stability, dimensional stability, and heat resistance, epoxy resins are widely used in many fields, such as building materials, electronic appliances, aerospace, coatings, and binders [1,2]. However, it is extremely flammable and releases a lot of heat and toxic fumes during combustion; this can seriously endanger the safety of human life and property [3,4] and also greatly limit its deeper application. Therefore, the flame-retardant modification of EP was an urgent problem for researchers to solve. In the past few decades, traditional brominated EPs have been widely used as the main flame retardant EPs due to their excellent cost performance. Although traditional halogenated flame retardants are economical and efficient, they not only emit a large amount of smoke when burned but also produce corrosive gases, seriously endangering human and environmental safety [3,4]. With the rapid increase in demand for high-efficiency flame retardants and halogen-free and environmentally friendly EP materials, new high-efficiency and environmentally friendly flame retardants have become a hotspot in the current research field.

Our laboratory has previously prepared Tris-(3-DOPO-1-propyl)-triazinetrione (TAD) [5], which improved the flame inhibition behavior of EPs with high LOI values and UL94 V-0 rating due to its excellent quenching and gas dilution effect. At the same time, it was found that TAD was not effective in suppressing heat and smoke release, especially the smoke, which contained toxic gases such as CO arising from incomplete combustion, which were even aggrandized during combustion. As we all know, in a fire, the first factor causing death is suffocation caused by toxic smoke and poisonous gas; therefore, although good flammability was achieved, measures still need to be taken to reduce the release of hot smoke, and especially the release of toxic gases [6].

Enhancing carbonization and forming a protective carbon layer to prevent heat transfer and mass exchange is expected to be a means to achieve this. Charcoal formation and smoke release are inversely proportional; therefore, improving the charcoal-forming capacity of the compound may become an effective solution for smoke suppression. For carbonization, catalytic carbonization of metal elements is one of the methods used, but metal elements exist in the form of metal ions, metal oxides, etc., and these will cause compatibility problems with organic substrates. Numerous studies have shown that organometallic frameworks could suppress the emission of toxic fumes. Most importantly, they catalyze themselves and the matrix, forming a protective carbon layer that prevents further exchange of heat and oxygen. For example, Wu et al. [7] synthesized a novel organic–inorganic hybrid carbonizer (SCTCFA-ZnO) that was combined with ammonium polyphosphate (APP) to prepare intumescent flame retardant polypropylene (PP) composites. The results showed that the excellent flame retardant performance and thermal stability performance were mainly attributed to the graphite crosslinked structure containing more P, O, N, and Si in the PP/APP/SCTCFA-ZnO, which promoted the formation of dense and continuous expanded graphite coke. Ma et al. [8] prepared organic triazine carbonization agent hybridized with zinc oxide (OTCA@ZnO), which was used to improve the flame retardant and smoke suppression properties of EVA OTCA@ZnO and exhibited excellent thermal stability after hybridization. Zheng et al. [9] prepared three different types of metal–organic frameworks (Co-MOF, Zn-MOF, and Fe-MOF) by self-assembly of organic bridging ligands and transition metal ions. The results revealed that excellent flame retardant performance was mainly attributed to the adsorption and catalytic oxidation of the metal–organic frameworks and the combination of catalytic carbonization of transition metal oxides. Hou et al. [10] synthesized a phosphorus-containing cobalt-based metal–organic framework by simple hydrothermal reaction for the fire safety of high-improvement EPs. Other methods of flame retardant include, for example, Bao et al. [11] grafted PMDA onto the surface of CNTs to prepare epoxy resin/CNTs-PMDA nanocomposites. Bekeshev et al. [12] added amphibole to epoxy composites, resulting in a decrease in the flammability of epoxy composites. Jian et al. [13] synthesized laminated phosphorus-based zinc triazole complexes (Zn-PT), which physically modified the epoxy resin to improve the flame retardancy of EPs. Liu et al. [14] synthesized a novel nitrogen-rich phosphoramidite D-5-AT for the modification of EPs.

Phosphazene has an alternating framework of P and N elements, which, as a unique phosphorus and nitrogen hybrid six-membered ring conjugated structure, has good thermal stability and flame retardancy [15,16,17]. Yang et al. [18] synthesized a phosphazene-based flame retardant (PBFA) containing an active amine group, which had good compatibility with the EP matrix. Zhang et al. then [19] chelated nickel ions (Ni^2+^) onto the PBFA surface (named PBFA-Ni^2+^). Fang et al. [20] prepared a polyphosphazene covalent triazine polymer (PCTP), and the LOI value of the EP composites increased significantly from 22.3% to 28.0%. All of the above passed the UL 94 V-0 rating and improved the EPs’ flame retardant and smoke suppression performance. As a coordination center, metal ions have the ability to promote the catalytic formation of carbon in composite materials during combustion, and the dense carbon layer formed can effectively isolate oxygen, flue gases, and heat [21,22]. Most of the visible smoke produced by combustion is mainly composed of toxic benzene derivatives, which metals can catalyze into more stable carbon materials or aromatic species, thereby inhibiting the production of smoke [23,24,25]. Studies have shown that organic ligands and metal ions both have structural characteristics of organic materials and inorganic materials through self-assembly; there is a strong interaction between organic ligands and polymer molecular chains, which greatly improves the compatibility and interfacial adhesion between flame retardant molecules and matrix materials. Moreover, organic ligands have a good dispersion in the matrix, which can significantly reduce the amount of flame retardant added [26,27]. The excellent thermal, smoke and carbon-forming properties of organometallic complexes make the flame retardant EP system have a good flame retardant effect, especially in the condensed phase, but there is still room for improvement in the gas phase [28,29,30].

To suppress the heat/smoke release of EP/TAD systems, an organozinc coordination compound based on phosphazene was designed and synthesized. Phosphorus elements in organic zinc complexes promoted carbon formation in EP composites [31]. The thermal properties, combustion behavior, and flame retardant action of the combination of Zn-PDH and TAD were studied. The system achieved a balance between the gas-phase and condense-phase action in its flame retardant effects, and the 3%TAD/1%Zn-PDH/EP composite system not only achieved good flame inhibition but also achieved a good smoke and heat suppression performance, showing a comprehensive flame retardant performance.

## 2. Results and Discussion

### 2.1. Structure Characterization

From Figure 1a, For HCCP, the characteristic peaks were P=N, P-N at 1190 cm^−1^ and 875 cm^−1^, and P-Cl absorption peaks at 525 cm^−1^ and 612 cm^−1^. For 4-APD, the peaks in the range of 3374–3450 cm^−1^ and 3304–3418 cm^−1^ belonged to the stretching vibration of N-H in them, while 1612–1647 cm^−1^ was the bending vibration of N-H. The absorption peak at 3092–3175 cm^−1^ was from the stretching vibration of C-H on the pyridine ring, and that between 750–1050 cm^−1^ was associated with the bending vibration of C-H. The absorption peaks of C=N and C-N corresponded to 1487–1508 cm^−1^ and 1283–1295 cm^−1^, respectively. For Zn-PDH, it was noteworthy that the stretching vibration of N-H changed from a double peak to a single peak, which was evidence of the change of the amino group to an imino group. Moreover, the characteristic peaks of HCCP and 4-APD were retained in Zn-PDH, while the P-Cl absorption peaks at 525 cm^−1^ and 612 cm^−1^ disappeared, which tentatively proved the synthesis of Zn-PDH. The infrared spectra of Zn-PDH showed distinct absorption peaks belonging to Zn-N at around 450 cm^−1^ while retaining the characteristic groups of the organoligands, which tentatively verified the successful synthesis of organometallic complexes.

The microscopic topography of Zn-4APD and Zn-PDH were observed by SEM, as shown in Figure 1c. Zn-4APD presents a rough surface with a large number of spheroid structures clustered together, and the size of the microspheres was basically maintained below 1μm. Zn-PDH shows a polyhedral structure of different sizes, all below 10 μm in size and irregular in shape. EDS test results before and after phosphazene modification are listed in Table 1. It was clear that phosphorus elements not contained before modification appeared in Zn-PDH, indicating successful HCCP access. In addition, the metal content of organometallic complexes was tested using ICP-MS, and the results were listed in Table 1. The significant reduction of metal content after phosphonitrile modification laterally confirms that HCCP has been successfully incorporated into organometallic complexes. The thermal properties of the synthesized organometallic complexes were analyzed by thermogravimetric testing, and the thermogravimetric curve is shown in Figure 1b. The results show that Zn-4APD has 96.8% residual carbon at 700 °C, and there was almost no weightlessness. The amount of residual carbon decreased significantly after modification with HCCP due to the change in the ratio between easily decomposable organic matter and non-decomposable metal elements.

### 2.2. Thermal Performance

The thermal degradation process of EP composites under N_2_ was studied by thermogravimetry; the TG and DTG curve were shown in Figure 2, and the relevant data were listed in Table 2. The initial degradation temperature of pure EP was 380.2 °C with a residue yield of 19.3 wt.% at 700 °C. After adding flame retardants, T_d,1%_ values of composites were decreased to different degrees. The results showed that all the flame retardant systems could reduce the initial decomposition temperature of EP composites and increase their carbon-forming capacity, maybe due to the promotion of the matrix decomposition by additives. In particular, the addition of the 4 wt.% of Zn-PDH resulted in a significant reduction of 113.7 °C for the T_d,1%_ of composites compared with pure EP, and the reduction of 4%TAD/EP was only 32.5 °C. It shows that Zn-PDH and TAD could promote the early decomposition of the epoxy matrix to varying degrees, and at the same time, confer thermal stability on EP composites at high temperatures. Especially when the higher proportion of Zn-PDH in the flame retardant system was higher, the lower the initial decomposition temperature of the EP composite was, and the higher the residual carbon content. This indicated that Zn-PDH has a more significant effect on matrix decomposition and catalytic carbonization, especially at lower temperatures. Moreover, it was noted that 4 wt.% Zn-PDH could increase the residual carbon content of pure EP by 80.3% at 700 °C, and 4 wt.% TAD could increase the residual carbon content of EP by 30.6%, which indicated the appropriate system presented better char-forming abilities at high temperatures. In addition to the good catalytic charring of EP by zinc ions, the increase in char yield may also be on account of the accelerated dehydration and carbonization reactions of EP by strong acids such as phosphoric and metaphosphoric acid produced by the thermal decomposition of organic ligand [32].

### 2.3. Flammability

The flame retardancy of EP composites was evaluated by LOI and UL 94 tests. As presented in Table 3, the LOI value of the pure EP sample was only 26.2%. With the addition of 4 wt.% TAD, the LOI value of EP reached 33.4%, and the vertical combustion passed the V-0 grade. This was because the phosphorous free radical released by TAD during combustion quenches the hydrocarbon free radical and terminates the combustion chain reaction in the gas phase [5]. On the other hand, during the decomposition of TAD, a large amount of inert nitrogen-containing gas was released, which dilutes the concentration of oxygen and the combustible gas released by the substrate, and further enhances the vapor flame retardation effect of TAD. The gas-phase flame retardancy of these phosphorous free radicals and inert nitrogen gases effectively improves the flame retardancy of the EP composites [33]. With the addition of 4 wt.% Zn-PDH, the LOI value of the EP reached 27% and reached no rating (NR) in the UL94 test. In all compounding systems, only 3%TAD/1%Zn-PDH/EP reached a V-0 level of vertical combustion, and the LOI number was as high as 33.2%. The results showed that the flame retardant effect of Zn-PDH was greatly improved after the addition of TAD, with an excellent gas-phase flame retardant effect and a flame retardant EP system with an excellent flame retardant performance can be obtained.

### 2.4. Combustion Behavior Analysis

The cone calorimeter was used to test the combustion performance of EP composites in a real fire environment, and the typical data and curves obtained were shown in Table 4 and Figure 3. As shown in Table 4, the ignition time of pure EP was 58 s. With the addition of flame retardants, the TTI of the EP composites was greatly shortened, which indicates that organometallic complexes can catalyze the early decomposition of the matrix to release flammable substances. The HRR curve in Figure 4a also shows that the peak heat release rate of flame retardant EP samples occurred about 20 s earlier than that of pure EP samples because the release of flammable substances caused the combustion reaction to proceed earlier. In particular, the inhibition effect of 1%TAD/3%Zn-PDH/EP was the most significant, which can reduce the PHRR and THR of EP by 61% and THR by 26%, respectively. From the total smoke release (TSR) curve of flame retardant EP in Figure 4c, it can be seen that the TSR value of EP decreased significantly with the addition of the flame retardant, and the smoke suppression ability of flame retardant EP samples showed an upward trend with the flame retardant gradual increase of the amount added. The TSR of 1%TAD/3%Zn-PDH/EP decreased by about 13% compared with pure EP, indicating that the introduction of metal ions has reached the expected goal of suppressing smog. Compared with EP, the introduction of phosphorus in 1%TAD/3%Zn-PDH/EP promotes the formation of dense phosphorus-rich carbon residue in the matrix, locking more matrix fragments in the condensed phase and inhibiting the release of heat and smoke gases. The av-EHC of 1%TAD/3%Zn-PDH/EP decreased by 15% than that of neat EP, which was because phosphorus-containing radicals from Zn-PDH pyrolysis brought inhibition or termination of free radical chain reactions of matrix [34,35], and the reduction of av-EHC values of the thermosets with TAD should be caused by the quenching effect of decomposed TAD fragments, which terminated the combustion-free radical chain reaction and decreased the combustion ratios of fuels. The reduced av-CO_2_Y value compared to pure EP indicated that the addition of flame retardants results in incomplete combustion, and the side indicates that the composite was largely retained in the condensed phase during combustion. As can be seen from the mass loss curve of Figure 4d, flame retardant EP, when added at 1%TAD/3%Zn-PDH, could increase the residual carbon content of the system by 162%, which was the highest residual carbon of any formulation. Figure 4d mass loss curve can also clearly show the obvious promotion effect of metal zinc and TAD on carbon-forming capacity. Table 5 showed that the presence of Zn-PDH was the main factor leading to the advanced decomposition of EP matrix. It can also be clearly observed from Figure 4a that 4%Zn-PDH/EP first appears as a minimal heat release peak around 50 s. This was precisely because the organic zinc complex promotes the decomposition of the matrix in advance to release combustible substances, and the subsequent rapid formation of the expanded carbon layer effectively inhibits the combustion intensity. With the continuous combustion reaction, the HRR of the barrier carbon layer with a certain strength was always maintained at a low state and finally reached the PHRR around 170 s. Compared with pure EP, the occurrence of the PHRR of 1%TAD/3%Zn-PDH/EP was delayed by 27 s and decreased by 61%, the carbon residue was increased by 162%, and the TSR was decreased by about 13%. Compared with EP, the PHRR of 3%TAD/1%Zn-PDH/EP was decreased by 49.3%, and the residual carbon was increased by 133.9%, which combined the advantages of 4%Zn-PDH/EP and 4%TAD/EP. In conclusion, when Zn-PDH and TAD are added to EP in appropriate proportions, the flame-retardant EP system obtained has excellent gas phase dilution and quenching effects and high carbon formation ability of the condensed phase.

Different flame retardant effects will affect the different combustion behavior of flame retardant materials. The difference between flame-retardant materials and pure samples when burned can be recorded by a conical calorimetric test. To more distinctly explain the action route of flame retardant effect, the three main modes of flame retardant action of the flame retardant materials were quantitatively evaluated. Three formulas [36] for calculating different flame retardant effects are shown in Equations (1)–(3), and the results were presented in Table 5.
(1)$$\mathrm{Flame}\;\mathrm{inhibition}\;\mathrm{effect}=\left(1-\frac{EHC_{fr}}{EHC_{pure}}\right)\times 100\%\;$$
(2)$$\mathrm{Charring}\;\mathrm{effect}=\left(1-\frac{TML_{fr}}{TML_{pure}}\right)\times 100\%\;$$
(3)$$\mathrm{Barrier}\;\mathrm{and}\;\mathrm{protective}\;\mathrm{effect}=\left(1-\frac{\frac{PHRR_{fr}}{PHRR_{pure}}}{\frac{THR_{fr}}{THR_{pure}}}\right)\times 100\%\;$$

As shown in Table 5, the flame retardant effect of 1%Zn-PDH/3%TAD/EP (12.3%) was better than that of 4%TAD/EP (7.78%). The carbonization effect increased with the increase of Zn-PDH addition, and a better barrier and protection effect was obtained. It shows that the addition of Zn-PDH improves the carbon-forming effect of the condensed phase because the excellent catalytic carbon-forming effect of Zn-PDH made the EP composite rapidly form a hard and dense carbon layer at a high temperature. It can be seen that the barrier and protective effect of charcoal depend not only on the number of residues but also on their quality, perhaps density [37,38]. The higher the amount of Zn-PDH added, the higher the P proportion is locked in the condensed phase, because the excellent catalytic carbon-forming effect of Zn-PDH made the EP composite rapidly form a hard and dense carbon layer at high temperatures, which limited the further separation of the matrix. Compared with 4%TAD, the flame retardant effect of 3%TAD/1%Zn-PDH/EP was better in the condensed phase.

### 2.5. Morphology Analysis of Residual Carbon

In order to further study the carbon layer quality of flame retardant EP composites, the macroscopic and microscopic morphology of residual carbon after the cone calorimeter test was observed. As shown in Figure 4, pure EP samples exhibit a loosely broken morphology with virtually no residual carbon. From the electron microscopy, it can be observed that there are many cruciform holes on the surface of the pure EP residual carbon because it does not have the ability to form a strong and dense carbon layer on its own. The key factor determining the flame retardant properties of a material is not only the amount of residual carbon but also the quality of the final residual carbon [39]. Although the addition of 4 wt.% TAD improved the carbon-forming ability of EP, there were still a lot of holes on its surface. This may be because the carbon layer generated during combustion was not strong enough to resist the release of phosphorous free radicals and inert nitrogen gas in TAD [40]. With the increase of Zn-PDH addition, the more the amount of residual, the higher the integrity and compactness of the carbon layer. However, a too dense carbon layer prevents further decomposition of the matrix and also cuts off the release channel of the phosphorus-containing free radicals. In particular, when TAD and Zn-PDH are added to EP in a ratio of 3:1, the flame retardant effect of the gas phase and condensed phase can be balanced to achieve a better flame retardant performance.

**Figure 4 molecules-28-03069-f004:**
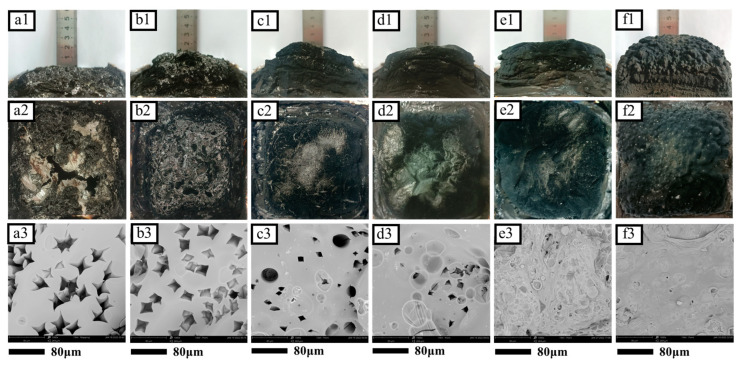
Digital photograph and SEM of carbon residue after cone calorimetry test of EP composite (×1000). EP (**a1**,**a2**,**a3**), 4%TAD/EP (**b1**,**b2**,**b3**), 3%TAD/1%Zn-PDH/EP (**c1**,**c2**,**c3**), 2%TAD/2%Zn-PDH/EP (**d1**,**d2**,**d3**), 1%TAD/3%Zn-PDH/EP (**e1**,**e2**,**e3**), and 4%Zn-PDH/EP (**f1**,**f2**,**f3**).

### 2.6. Analysis of Gas Phase Mechanism

To further explore the gas-phase flame retardancy of the flame retardant EP system, TG-FTIR tests were conducted on EP and 3%TAD/1%Zn-PDH/EP, and the test results were shown in Figure 5. At 350 °C, the 3%TAD/1%Zn-PDH/EP sample has lysed a small amount of fragment structure containing P=O (1258 cm^−1^) and phenolic compounds (CAr-O, 1174 cm^−1^). This was caused by the combination of phosphor-containing free radicals and phenoxy free radicals released by TAD decomposition with alkyl free radicals and hydrogen free radicals generated by EP decomposition. With the increase in temperature, phosphor-containing free radicals and phenoxy free radicals are released rapidly, which creates a good gas phase quenching effect. Additionally, it can be observed in the A region of Figure 5A that EP releases a large number of carbonyl compounds (1664–1810 cm^−1^) during the combustion process, while in Figure 5B, the carbonyl compounds released by 3%TAD/1%Zn-PDH/EP are significantly reduced. In addition, there were more benzene ring structures (C=C, 1607 cm^−1^) in the pyrolysis products of 3%TAD/1%Zn-PDH/EP; this was because the Zn in Zn-PDH promoted the decomposition of polymer macromolecules to a certain extent and produced more aromatic structures, which reduced the release of combustible substances in the combustion process of EP materials. Therefore, TAD and Zn-PDH played a good synergistic flame retardant effect.

### 2.7. Analysis of Condensed Phase Mechanism

In order to observe the carbon formation behavior of EP composites in the thermal degradation process, the EP composites were placed in a Muffle furnace and stored at different temperatures for 15 min. The digital photos were shown in Figure 6. As can be seen from the figure, the expansion process of pure EP was mainly around 350 °C to 450 °C, the expansion shape was not controlled, and the expansion volume reached more than three times its own volume. As the temperature continued to rise, the residual carbon of the EP gradually decreased and completely disappeared at about 700 °C. On the contrary, the expansion volume of 4%Zn-PDH/EP during thermal decomposition was almost negligible, and a large amount of residual carbon was still retained at 700 °C. It was speculated that due to the excellent catalytic carbon formation of Zn-PDH, a hard and compact carbon layer was rapidly formed in the epoxy resin composite at high temperatures, which restricted the further decomposition of the matrix. During the combustion of Zn-PDH, a high-quality carbon layer containing a P-O-C crosslinked structure can be generated, which reduces the release of heat and smoke toxic gas by retaining more matrix fragments in the residual carbon. The expansion temperature range of 4%TAD/EP is similar to that of pure EP, but the expansion shape and volume of TAD/EP are limited due to the formation of a carbon layer with a certain strength, and most of the residual carbon can be retained at about 700 °C. The carbon forming capacity of 3%TAD/1%Zn-PDH/EP is between the two, and the solid carbon layer also limits the expansion volume. However, because the phosphoric acid substances and metal oxides produced by metal elements and phosphorus elements during the combustion process jointly cover the surface of the matrix, preventing the decomposition of the matrix and the release of combustible components, inhibiting the combustion reaction and rapidly forming a dense carbon layer, effectively inhibiting the release of heat.

To further investigate the mechanism of action of flame retardants in polymers, FTIR tests were performed on samples at each temperature in a Muffle furnace, as shown in Figure 7. There were N-H (3470 cm^−1^ and 1620 cm^−1^), aliphatic -CH_3_ (2950 cm^−1^) and -CH_2_ (2860 cm^−1^), C-O-C (1265–1010 cm^−1^), and C=O (1740 cm^−1^) C-N (1320 cm^−1^) and benzene ring (3040 cm^−1^, 1510–1450 cm^−1^, 815 cm^−1^ and 740 cm^−1^) in pure EP and flame retardant EP. The biggest difference between the two flame retardant systems is that the characteristic peaks of P=O (1250–1260 cm^−1^) and P-O-C (919–983 cm^−1^) can be obviously observed in 3%TAD/1%Zn-PDH/EP, which helps to form the crosslinked carbon layer. At the same time, the N-H content should actually be increased after the addition of flame retardant, but the N-H content in 3%TAD/1%Zn-PDH/EP was relatively lower than that in pure EP. This was because flame retardants interact with each other, allowing more element N to enter the gas phase in the form of inert gas, diluting the concentration of combustible fuels and helping to suppress the combustion reaction. Compared with pure EP, there were still a large number of benzene ring structures at high temperatures in the 3%TAD/1%Zn-PDH/EP, indicating that the addition of flame retardants delayed the degradation of matrix. Seemingly contrary to the results of TG-FTIR, in fact, this was because the flame retardant mainly promoted the decomposition of the matrix into carbon before 400 °C. At this time, the matrix was released into the air in the form of large molecular fragments, reducing the release of combustible substances. With the increase in temperature, the flame retardant promotes the formation of the P-O-C crosslinked reinforced carbon layer in the matrix, which keeps more matrix fragments in the cohesive phase and inhibits the further decomposition of the matrix, thus achieving a good flame retardant effect.

XPS was used to study the surface structure and composition of conical residual carbon of EP composites, in which the element content and structure are shown in Table 6 and Figure 8. It can be seen that most of the carbon elements in 3%TAD/1%Zn-PDH/EP exist in the form of C=C and C-C (284.8 eV), indicating that most of the matrix structure was retained in the residual carbon. For N1s, most nitrogen elements to C=N (400.2 eV) and C-N, P-N (398.6 eV) structure exists, conducive to strengthening the strength of the carbon layer. The low content of N-H was because most of the N-H was released into the gas phase as NH_3_, diluting the flammable gas. Most of the oxygen elements of 3%TAD/1%Zn-PDH/EP exist in the stable crosslinked structure of C-O-C or C-O-P (532.8 eV), and the rest exist in the structure of P=O or C=O (531.0 eV) in the carbonyl group, which also contributes to the formation of the crosslinked carbon layer. The fitting curve of P2p includes P=O (134.8 eV), P=N and P-O (133.93 eV), and P-N (133.13 eV) structures. In conclusion, 3%TAD/1%Zn-PDH/EP system could promote the formation of crosslinked carbon layer containing P and N elements and release inert gas to dilute flammable volatiles, thus inhibiting the combustion reaction.

In addition, the reserved and released phosphorus (P) contents were calculated and shown in Table 7. From Table 7, for 4%TAD/EP, most of P (79%) was released into the gas phase, whereas for the samples with only Zn-PDH, all P was basically retained in the condensed phase. This indicated that P-containing compounds or species, such as phosphoric acid, pyrophosphoric acid, polyphosphoric acid, and so on, from phosphonitrile, more effectively participated in the charring reactions with the decomposition products to form P-rich residues due to the catalytic charring effect of zinc ions. TAD is mainly released into the gas phase as P-containing free radicals and plays the role of free radical quenching. However, combining TAD and Zn-PDH in EP balanced the distribution of phosphorus in the gas and condense phases, with half released and half retained. This indicated combination of TAD/Zn-PDH, endowed the EP with both the gas-phase and condense-phase flame retardant modes of action, reflected by its comprehensive flame retardant properties.

Raman spectroscopy was used to explore the graphitization degree of residual carbon in EP composites, as shown in Figure 9. Peak D (1360 cm^−1^) and peak G (1580 cm^−1^) correspond to the vibration of carbon atoms in the disordered and ordered graphite structures, respectively. The degree of graphitization (R) of residual carbon was evaluated by the ratio of the integrated peak area of the two bands of peak D and peak G. The lower the ratio, the higher the degree of graphitization, and the better the corresponding flame retardant performance. Raman spectra showed that the R-values of flame retardant EP were all lower than those of pure EP, indicating that the introduction of flame retardant promoted the graphitization of residual carbon and formed a higher-quality carbon layer. The highly graphitized carbon layer can effectively prevent further combustion of the matrix, improving the protective effect of the barrier and, thus, achieving more excellent flame retardant properties. Similar to the results observed in the SEM figure, the higher the amount of Zn-PDH added, the higher the graphitization degree of the residual carbon.

### 2.8. Speculation of Flame-Retardant Mechanism

As shown in Figure 10 in the gas phase, the fragmented structure was cracked out of the 3%TAD/1%Zn-PDH/EP sample, which was formed by combining the phosphorus-containing radicals and phenoxy radicals released by the decomposition of TAD and the alkyl radicals and hydrogen radicals produced by the decomposition of EP. With the temperature increase, phosphorus-containing and phenoxy radicals were rapidly released, which created a good gas-phase quenching effect. Flame retardants interacted with each other to allow more N elements to enter the gas phase as an inert gas, diluting the concentration of flammable fuels. A large number of carbonyl compounds were released during EP combustion, and the carbonyl compounds released by 3%TAD/1%Zn-PDH/EP were significantly reduced. The Zn in Zn-PDH promoted the decomposition of polymer macromolecules to a certain extent to produce more aromatic structures, which reduced the release of flammable substances in EP materials during combustion. Flame retardants mainly played a role in promoting the decomposition of the matrix into carbon, at which time the matrix was released into the air in the form of macromolecular fragments, reducing the release of flammable substances. In the condensed phase, with the increase of temperature, the flame retardant promotes the matrix to form a reinforced carbon layer rich in the P-O-C crosslinked structure, thereby retaining more matrix fragments in the condensed phase, inhibiting the further decomposition of the matrix and creating a good flame retardant effect.

## 3. Experimental

### 3.1. Materials

The organozinc complex Zn-PDH and Tri-(3-DOPO-1-propyl)-triazintrione (TAD) were made by the laboratory. [5] Bisphenol A glycidyl ether (EP, E-51) was purchased from Nantong Star Synthetic Materials Co., Ltd. (Nantong, China); 4,4-Diaminodiphenylmethane (DDM) was gained from Sinopharm Chemical Reagent Co., Ltd. (Shanghai, China). The physicochemical and technological properties of components were listed in Table 8.

### 3.2. Synthesis of Zn-PDH

Place zinc acetate dihydrate (8.78 g, 0.04 mol) in a 100 mL four-mouth flask to which add 4-aminopyridine (4-APD) (7.53 g, 0.08 mol) of 60 mL of methanol solution. Increase to reflux temperature and heat, stirring for 4 h. After the end of the reaction, the impurities are removed by filtration, the filtrate is volatilized in the air, and the obtained white crystalline precipitate is washed with water to remove impurities. Finally, it was dried in a blast oven at 110 °C to obtain the organozinc complex Zn-4APD (white powder).

Weigh the Zn-4APD (20 g, 0.045 mol) prepared above in a flask, add 300 mL of dried chlorobenzene, and increase the temperature to 110 °C. After dispersing evenly, triethylamine (22.923 g, 0.226 mol) was added. Hexachlorocyclotriphosphazene (HCCP) (28.35 g, 0.082 mol) was then dissolved in 200 mL of dried chlorobenzene, and the clarification solution is slowly added dropwise to the four-mouth flask using a constant pressure dripping funnel, and the timing reaction was 24 h after significant reflux was observed in the three-mouth flask. After the end of the reaction, it is allowed to cool to room temperature and then start filtration, and the obtained precipitate is washed with ethanol to remove impurities. Finally, it was dried in a blast oven at 80 °C to obtain the organic zinc complex Zn-PDH (reddish-brown solid). The reaction formula was shown in Figure 11.

### 3.3. Characterizations

#### 3.3.1. Fourier Transform-Infrared Spectroscopy (FTIR)

The FTIR spectroscopy was performed by the KBr press method on a Nicolet iN10MX spectrometer from Thermo-Scientific, Waltham, MA, USA, in the wavelength range from 500 to 4000 cm^−1^.

#### 3.3.2. Thermogravimetry–Fourier Transform Infrared (TG-FTIR) Test

A thermogravimetry–Fourier transform infrared (TG-FTIR) test was carried out on an STA 8000 simultaneous thermal analyzer produced by Perkin Elmer. The sample was placed in the heat of alumina crucibles and heated from 50 to 700 °C at a heating rate of 20 °C/min under a nitrogen atmosphere. FTIR was used to collect sample pyrolysis gas fragments during TGA process.

#### 3.3.3. Limiting Oxygen Index (LOI) Measurement

The limited oxygen index tests were conducted using a 300800 LOI instrument produced by Concept Ltd., Swindon, UK. According to ASTM D2863-19, sample sizes were 130.0 mm × 6.5 mm × 3.2 mm.

#### 3.3.4. UL94 Vertical Combustion Test

The flame-retardant rating of flame-retardant EP was obtained by vertical combustion test using an instrument FTT0802 type combustion test chamber produced by Fire Testing Technology Ltd., East Grinstead, UK. According to ASTM D2863-19, sample sizes were 125.0 mm × 12.7 mm × 3.2 mm.

#### 3.3.5. Cone Calorimeter Test (CONE)

According to the ISO5660 standard, the FTT0007 cone calorimeter from the British company was used under the condition of an external heat of 50 kW/m^2^. Sample sizes were 100.0 mm × 100.0 mm × 3.2 mm. Samples of each recipe were tested twice.

#### 3.3.6. Scanning Electron Microscopy (SEM) Test

We used the Phenom Pro scanning electron microscope (SEM) produced by the Dutch company Phenom™ World at vacuum conditions with a voltage of 10 kV. The test specimen was treated with gold spray at a current intensity of 30–35 mA for 180 s in an argon atmosphere.

#### 3.3.7. X-ray Photoelectron Spectrometer (XPS)

The element compositions of the residues were investigated using a Perkin Elmer PHI 5300 ESCA X-ray photoelectron spectrometer (XPS).

#### 3.3.8. Laser Raman Spectroscopy (LRS)

The LRS measurements were performed using LabRAM HR Evolution laser Raman spectroscopy (LRS) from Horiba in France, with a laser wavelength of 532 nm and a scanning range of 4000–500 cm^−1^.

### 3.4. Preparation of Epoxy Resin Composites

Under mechanical stirring, a certain quality of Zn-PDH and TAD were successively dispersed into the EP at 120 °C. After the flame retardant is completely dissolved and dispersed, the temperature is reduced to below 110 °C, DDM is added to the mixture and stirred until completely melted. After pumping the mixture in a vacuum oven at 120 °C for 3 min, it is quickly transferred to a preheated mold and placed in a blast oven, pre-cured at 120 °C for 2 h, and cured at 170 °C for 4 h. Prepared a control sample EP without flame retardants according to the method described above. The formulations of EP composites were listed in Table 9.

## 4. Conclusions

A novel flame-retardant Zn-PDH presented excellent thermal stability and charring ability. The 3%TAD/1%Zn-PDH/EP system had excellent thermal stability and charcoal-forming ability while achieving vertical combustion V-0 and LOI up to 33.2%. At the same time, the addition of the 3%TAD/1%Zn-PDH system to EP promoted the decomposition of the epoxy matrix in advance so that EP composites had thermal stability at high temperatures. Compared with 4%TAD/EP, 3%TAD/1%Zn-PDH/EP system not only inhibited the inflammability of the EP composites but also obtained good smoke and heat suppression properties. In the gas phase, the phosphorus-containing radicals and benzene-oxygen radicals released by the decomposition of TAD combined with the alkyl radicals and hydrogen radicals produced by the decomposition of EP to form a fragmented structure and crack out, and with the increase in temperature, phosphorus-containing radicals, and phenoxy radicals were rapidly released, which exerted a good gas phase quenching effect. At the same time, the excellent catalytic carbonization of Zn-PDH enabled EP composites to rapidly form a hard and dense carbon layer at high temperatures, which limited the further decomposition of the matrix. Further, it allowed more N elements to enter the gas phase as an inert gas, diluting the concentration of flammable fuels and helping to inhibit the combustion reaction. TAD/Zn-PDH system balanced the flame retardant effect of the gas phase and the condensed phase, resulting in comprehensive flame retardant performance.

## Figures and Tables

**Figure 1 molecules-28-03069-f001:**
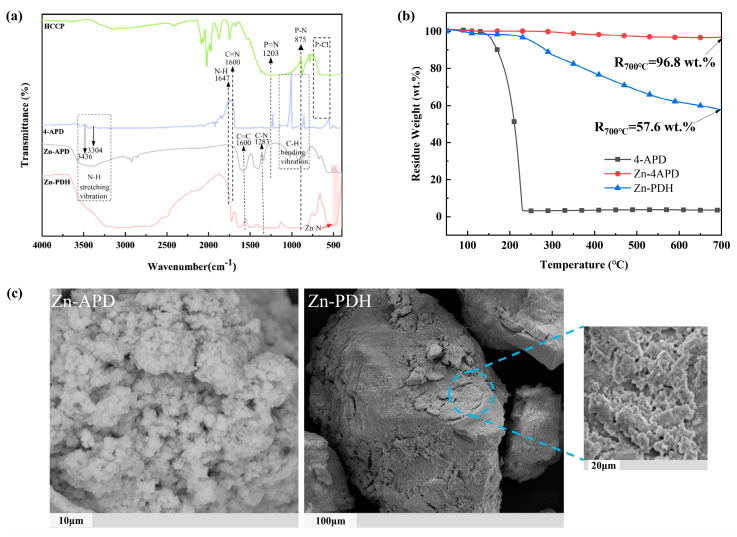
FTIR of flame retardants (**a**), the TGA curves of intermediate and final products (**b**), and SEM photos of the Zn−APD and Zn−4APD (**c**).

**Figure 2 molecules-28-03069-f002:**
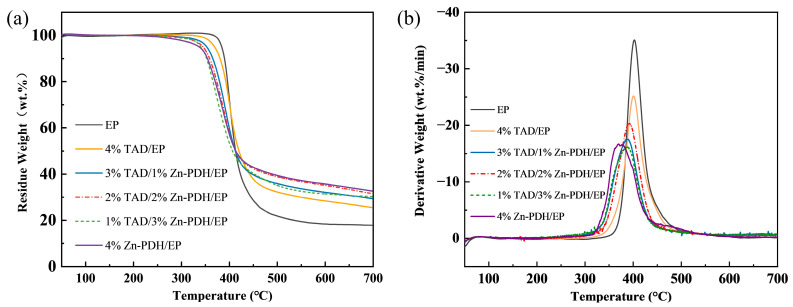
TGA (**a**) and DTG (**b**) curves of EP composites.

**Figure 3 molecules-28-03069-f003:**
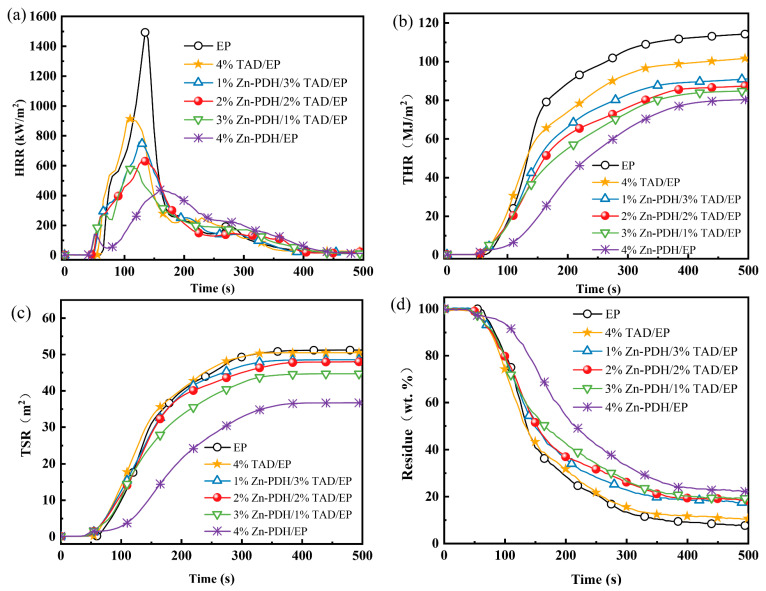
Curve of HRR (**a**), THR (**b**), TSR (**c**), and residue (**d**) of flame-retardant EP.

**Figure 5 molecules-28-03069-f005:**
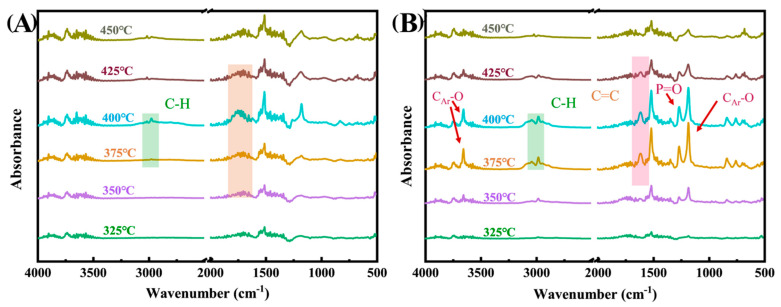
TG-FTIR curves of EP composites: EP (**A**), 3%TAD/1%Zn-PDH/EP (**B**).

**Figure 6 molecules-28-03069-f006:**
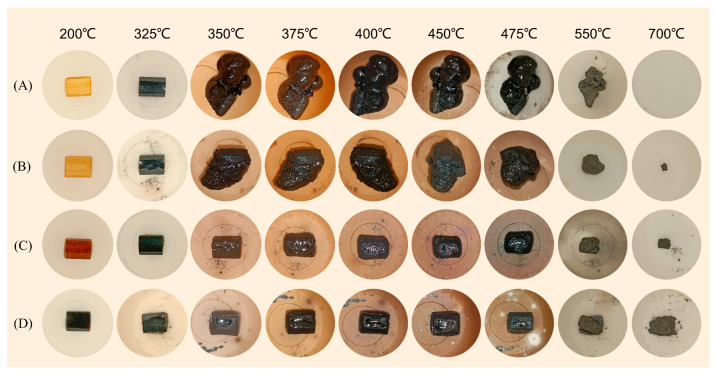
Digital photographs of EP (**A**), 4%TAD/EP (**B**), 3%TAD/1%Zn-PDH/EP (**C**), 4%Zn-PDH/EP (**D**) after being maintained at different temperatures for 15 min in a muffle furnace.

**Figure 7 molecules-28-03069-f007:**
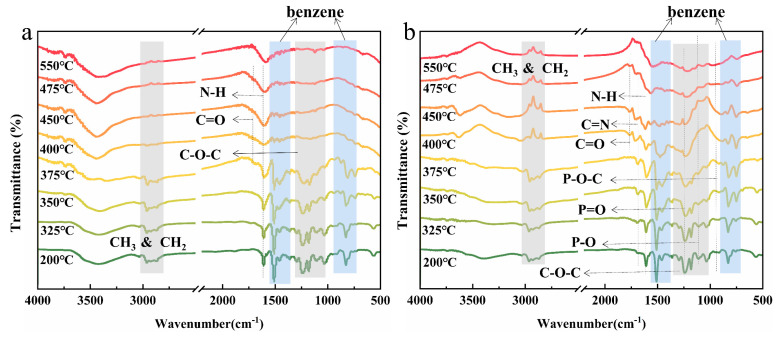
Infrared spectra of EP (**a**) and 3%TAD/1%Zn-PDH/EP (**b**) after being maintained at different temperatures for 15 min in a muffle furnace.

**Figure 8 molecules-28-03069-f008:**
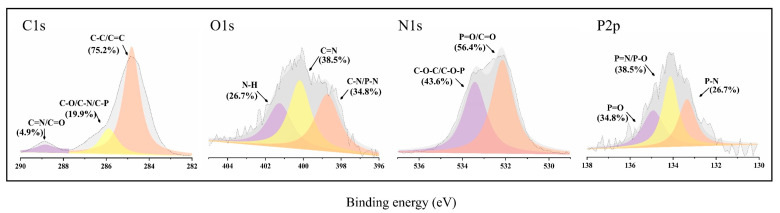
XPS spectra of residual carbon after cone size test of EP composites: C1s, O1s, N1s, and P2p.

**Figure 9 molecules-28-03069-f009:**
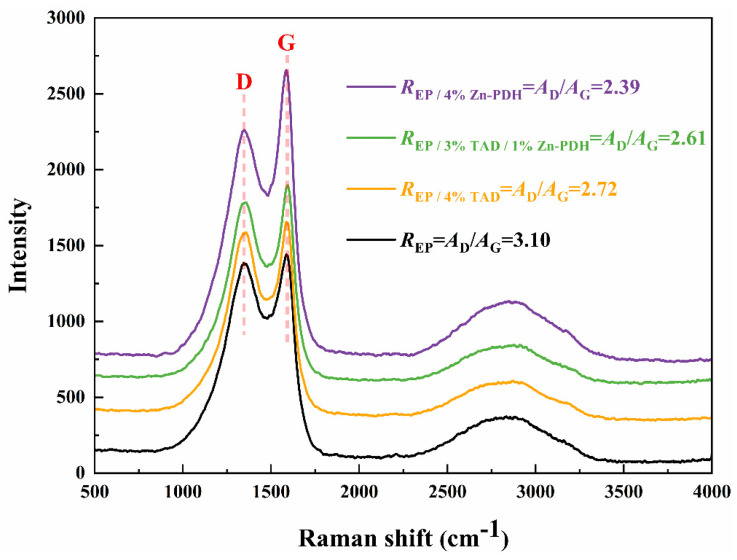
Raman spectra of epoxy resin composites.

**Figure 10 molecules-28-03069-f010:**
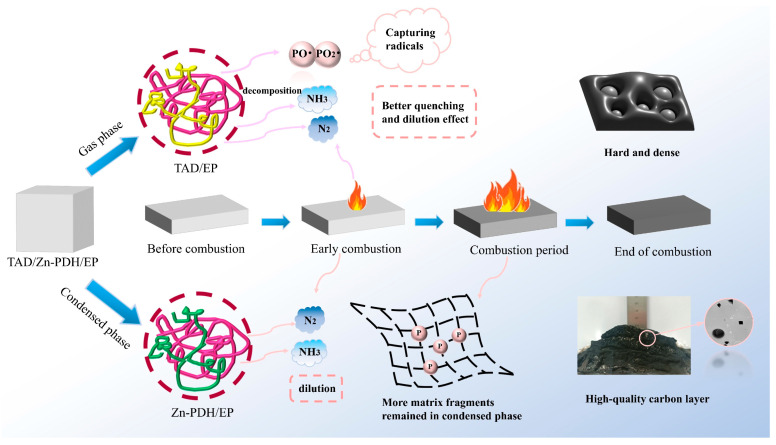
Flame retardant modes of action of TAD/Zn-PDH in EP.

**Figure 11 molecules-28-03069-f011:**
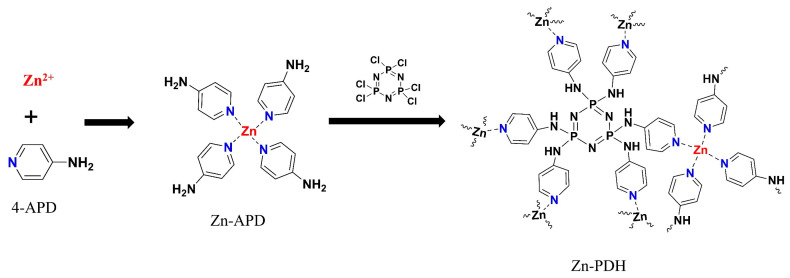
Synthetic route of Zn-PDH.

**Table 1 molecules-28-03069-t001:** EDS data for organometallic complexes.

Sample	Element Sample	Atomic Conc.	Weight Conc.
Zn-4APD	O	41.83	27.74
	C	38.61	19.22
	Zn	19.57	53.04 (58.02) ^a^
Zn-PDH	C	43.65	34.54
	O	25.57	26.96
	N	23.07	21.29
	P	6.19	12.62
	Cl	0.98	2.29
	Zn	0.53	2.30 (25.3) ^a^

^a^ Content of metal elements in organic zinc complexes tested by ICP.

**Table 2 molecules-28-03069-t002:** TGA of EP composites under N_2_ conditions.

Sample	T_d,1%_ (°C)	T_peak_ (°C)	Amount of Residual Carbon (wt.%)		
			500 °C	600 °C	700 °C
EP	380.2	400.7	23.4	19.9	19.3
4%TAD/EP	347.7	400.6	32.1	28.3	25.2
3%TAD/1%Zn-PDH/EP	311.7	392.2	35.8	32	29.3
2%TAD/2%Zn-PDH/EP	300.4	387.9	38.7	35	31.4
1%TAD/3%Zn-PDH/EP	290.4	387.3	35	31.3	30.1
4%Zn-PDH/EP	266.5	369.7	41.9	38	34.8

**Table 3 molecules-28-03069-t003:** LOI values and UL94 levels of epoxy resin composites.

Sample	LOI [%]	UL94	
		Drip	Level
EP	26.2	Yes	No
4%TAD/EP	33.4	No	V-0
3%TAD/1%Zn-PDH/EP	33.2	No	V-0
2%TAD/2%Zn-PDH/EP	31.3	No	V-1
1%TAD/3%Zn-PDH/EP	29.0	No	V-1
4%Zn-PDH/EP	27.0	No	No

**Table 4 molecules-28-03069-t004:** Parameters of cone calorimetry test.

Sample	TTI [s]	PHRR [kW m^−2^]	THR [MJ m^−2^]	av-EHC [MJ kg]	av-COY [kg kg^−1^]	av-CO_2_Y [kg kg^−1^]	TSR [m^2^/m^2^]	Residue [wt.%]
EP	58	1492	114.30	27.52	0.10	2.19	5781	7.28
4%TAD/EP	52	912	101.60	25.38	0.13	1.88	5702	10.50
3%TAD/1%Zn-PDH/EP	44	756	90.97	24.21	0.12	1.83	5485	17.03
2%TAD/2%Zn-PDH/EP	41	630	87.44	23.76	0.11	1.81	5418	18.22
1%TAD/3%Zn-PDH/EP	38	579	84.78	23.39	0.10	1.80	5045	19.07
4%Zn-PDH/EP	39	436	80.26	22.88	0.10	1.77	4138	22.06

**Table 5 molecules-28-03069-t005:** Quantitative assessment of three main flame-retardant modes of action.

Samples	Flame Inhibition Effect (%)	Charring Effect (%)	Barrier and Protective Effect (%)
Neat EP	-	-	-
4%TAD/EP	7.78	3.47	31.23
1%Zn-PDH/3%TAD/EP	12.03	10.52	36.34
2%Zn-PDH/2%TAD/EP	13.66	11.80	44.80
3%Zn-PDH/1%TAD/EP	15.00	12.72	47.69
4%Zn-PDH	16.86	16.03	58.39

**Table 6 molecules-28-03069-t006:** Carbon residue data of EP composites after cone size test from XPS.

Sample	C [wt.%]	O [wt.%]	N [wt.%]	P [wt.%]	Zn [wt.%]
4%TAD/EP	82.33	13.29	3.52	0.86	-
3%TAD/1%Zn-PDH/EP	83.28	15.78	0.73	1.13	0.12
4%Zn-PDH/EP	66.68	23.81	5.67	2.26	0.58

**Table 7 molecules-28-03069-t007:** Release before and after combustion and P content of the condensed phase.

Sample	P Ration in Residues (%)	Char Yield (%)	Initial P Ration in Samples (%)	Reserved P Ratio in Total P (%)	Released P Ratio in Total P (%)
4%TAD/EP	0.86	10.5	0.43	21.00	79.00
3%TAD/1%Zn-PDH/EP	1.13	17.03	0.39	49.34	50.66
4%Zn-PDH/EP	2.26	22.06	0.50	99.71	0.29

**Table 8 molecules-28-03069-t008:** Physicochemical and technological properties of components.

The Qualitative Characteristics	Value
epoxy resin E-51	
Epoxy functionalities	2.5~6.0
Viscosity, Pa × s	5
Density at 25 °C, g/cm^3^	1.22
Molecular weight	600
DDM	
Molecular weight	198.264
Density at 25 °C, g/cm^3^	1.15
Boiling temperature, °C	242
Zn-PDH	
Molecular mass, g/mol	1364.46
Density at 25 °C, g/cm^3^	0.975
Phosphorus content, % by weight	6.8
Decomposition temperature, °C	111.86
TAD	
Molecular mass, g/mol	861
Density at 25 °C, g/cm^3^	1.449
Tg, °C	113
Decomposition temperature, °C	291

**Table 9 molecules-28-03069-t009:** Formulations of epoxy resin composites.

Samples	EP (g)	DDM (g)	TAD (g)	Zn-PDH (g)
EP	100	25.3	--	--
4%TAD/EP	100	25.3	5.22	--
3%TAD/1%Zn-PDH/EP	100	25.3	3.92	1.31
2%TAD/2%Zn-PDH/EP	100	25.3	2.61	2.61
1%TAD/3%Zn-PDH/EP	100	25.3	1.31	3.92
4%Zn-PDH/EP	100	25.3	--	5.22

## Data Availability

The data presented in this study are available on request from the corresponding author. The data are not publicly available.

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
