# Peer review of "Study on Flame Retardancy Behavior of Epoxy Resin with Phosphaphenanthrene Triazine Compound and Organic Zinc Complexes Based on Phosphonitrile"

_molecules, 2023, doi:10.3390/molecules28073069_

Round 1

Reviewer 1 Report

In this work, the author investigated the thermal properties, combustion behavior and flame retardant action of the combination of Zn-PDH and TAD on the EP. This manuscript was well written. Therefore, I suggest accepting this paper after a minor revision. Before resubmitting, the authors should solve the following problems:

1. Why did the authors offer char residue with different temperature?

2. The 3%TAD/1%Zn-PDH/EP reached UL 94 V-0 rating; however, 2%TAD/2%Zn-PDH/EP UL 94 rating is V-1. Is there any reason behind it?

3. The introduction is lengthy, and the expression idea is not clear. Recent progress in flame retardant epoxy resin could be cited to enrich the introduction, such as Journal of Colloid and Interface Science, 2022, 609, 513-522, Polymer Degradation and Stability Volume 2022, 196, 109840. Please reconceive the introduction, which needs to show the reason and basis of the experimental idea.

4. The English language of the entire manuscript needs to be polished, and some grammatical errors should be corrected in time. Such as the synthetic part of Zn-PDH “... it is dried in a blast oven at 80 °C to obtain the organic zinc complex Zn-PDH (reddish-brown solid). The reaction formula is shown in Fig 1.” should be “were”.

5. Line 198, Remove the period before the “in pure EP and flame retardant EP”.

Author Response

Comment 1: Why did the authors offer char residue with different temperature?

Response: Thanks. Through keeping samples in a muffle furnace with regularly lifted temperatures for fixed time, the morphological changes of the residual char can be observed during thermal degradation. Then, the structural changes of residues at different temperatures can also be obtained through FTIR tests. This is a usual method used in previous literatures and it can give the morphological and structural information of char residues at different temperatures, helpful for illustrating the flame retardant mechanism in condense phase.

Comment 2: The 3%TAD/1%Zn-PDH/EP reached UL 94 V-0 rating; however, 2%TAD/2%Zn-PDH/EP UL 94 rating is V-1. Is there any reason behind it?

Response: Thank you for the reviewer's suggestion, the 2% content of Zn-PDH forms a relatively dense char layer at the early stage, which enables the gas to break through the char layer and thus blow out the flame after accumulating for a certain period of time. After the second ignition, the enhanced char layer prevented the internal gas from being ejected outward, and the pyrolysis products could not penetrate the solid char layer to form a blow-out effect. However, 1% Zn-PDH/EP formed a medium-strength char layer after combustion, which not only prevented the heat from diffusing to the internal matrix, but also could break through the char layer and blow out the flame when the volatiles accumulated to a certain amount, and had an obvious blow-out effect after both ignitions. Therefore, the flame-retardant effect of 2% Zn-PDH/EP is relatively better.

Comment 3: The introduction is lengthy, and the expression idea is not clear. Recent progress in flame retardant epoxy resin could be cited to enrich the introduction, such as Journal of Colloid and Interface Science, 2022, 609, 513-522, Polymer Degradation and Stability Volume 2022, 196, 109840. Please reconceive the introduction, which needs to show the reason and basis of the experimental idea.

Response: Thanks for the referee’s suggestion about the structure of introduction. As Reviewer suggested, We have streamlined the introduction and have cited the article and marked those in red in the revised manuscript.

Comment 4: The English language of the entire manuscript needs to be polished, and some grammatical errors should be corrected in time. Such as the synthetic part of Zn-PDH “... it is dried in a blast oven at 80 °C to obtain the organic zinc complex Zn-PDH (reddish-brown solid). The reaction formula is shown in Fig 1.” should be “were”.

Response: Thanks for the referee’s suggestion. As Reviewer suggested, we have completed the revisions and marked those in red in the revised manuscript.

Comment 5: Line 198, Remove the period before the “in pure EP and flame retardant EP”.

Response: Thanks for the referee’s suggestion. As Reviewer suggested, we have completed the revisions and marked those in red in the revised manuscript.

Reviewer 2 Report

The manuscript under the title: “Study on flame retardancy behavior of Epoxy resin with phosphaphenanthrene triazine compound and organic zinc complexes based on phosphonitrile” is in line with Molecules journal. This topic is relevant and will be of interest to the readers of the journal. It based on original research. This research has scientific novelty and practical significance. The article has a typical organization for research articles.
Before the publication it requires significant improvements, especially:

  1. The "Introduction" section: it has been proven that the effect of various modifying additives and fillers on the flammability reduction and physical and chemical properties of epoxy polymer composites is determined by many factors: ……. I think the related references should be cited corresponding to each aspect, e.g. (but not limited to these), which will undoubtedly improve the "Introduction" section:
  • Polymers 202113(15), 2421; https://doi.org/10.3390/polym13152421
  • Polymers 202113(19), 3332; https://doi.org/10.3390/polym13193332

·        Inorg. Mater. Appl. Res. 2019, 10, 1135–1139, https://doi.org/10.1134/S2075113319050228

·        Polymer Composites20204120252035https://doi.org/10.1002/pc.25517

  1. Also in the "Introduction" section, it is necessary to more clearly formulate the scientific novelty of this work and indicate how it differs from those available in the literature.
  2. Section 2.1. It is necessary to add the physicochemical characteristics of components - give a table with the main physicochemical and technological properties of epoxy resin, hardener and other components.
  3. To confirm the structure of the synthesized substance according to the method described in paragraph 2.2. it would be nice to present NMR and FTIR data.
  4. It is known that the introduction of dispersed fillers to an amount of more than 1 wt.% often leads to a decrease in the physical and mechanical characteristics of polymer composites. it would be nice to show how the filler you synthesized affects the strength of epoxy composites.

Author Response

Comment 1: The "Introduction" section: it has been proven that the effect of various modifying additives and fillers on the flammability reduction and physical and chemical properties of epoxy polymer composites is determined by many factors: ……. I think the related references should be cited corresponding to each aspect, e.g. (but not limited to these), which will undoubtedly improve the "Introduction" section:

Polymers 2021, 13(15), 2421; https://doi.org/10.3390/polym13152421

Polymers 2021, 13(19), 3332; https://doi.org/10.3390/polym13193332

  Inorg. Mater. Appl. Res. 2019, 10, 1135–1139, https://doi.org/10.1134/S2075113319050228

Polymer Composites. 2020; 41: 2025–2035. https://doi.org/10.1002/pc.25517

Response: Thanks for the referee’s suggestion about the structure of introduction. As Reviewer suggested, We have cited the article and marked those in red in the revised manuscript.

Comment 2: Also in the "Introduction" section, it is necessary to more clearly formulate the scientific novelty of this work and indicate how it differs from those available in the literature.

Response: Thanks for the referee’s suggestion about the structure of introduction. As Reviewer suggested, We have added the novelty and marked those in red in the revised manuscript.

Comment 3: Section 2.1. It is necessary to add the physicochemical characteristics of components - give a table with the main physicochemical and technological properties of epoxy resin, hardener and other components.

Response: Thanks for the referee’s suggestion. As Reviewer suggested, We have supplemented the table in section 2.1.

Comment 4: To confirm the structure of the synthesized substance according to the method described in paragraph 2.2. it would be nice to present NMR and FTIR data.

Response: Thanks for your precious suggestion. We added infrared and scanning electron microscopy. Tested by infrared spectroscopy, for Zn-PDH, it was noteworthy that the stretching vibration of N-H changes from a double peak to a single peak, which was evidence of the change of the amino group to an imino group. Moreover, the characteristic peaks of HCCP and 4-APD were retained in Zn-PDH, while the P-Cl absorption peaks at 525 cm-1 and 612 cm-1 disappeared, which tentatively proved the synthesis of Zn-PDH. The infrared spectra of Zn-PDH showed distinct absorption peaks belonging to Zn-N at around 450 cm-1 while retaining the characteristic groups of the organoligands, which tentatively verified the successful synthesis of organometallic complexes. In addition, Zn-PDH shows a polyhedral structure of different sizes, all below 10 μm in size and irregular in shape. EDS test results before and after phosphazene modification are listed in Table 3. It was clear that phosphorus elements not contained before modification appear in Zn-PDH, indicating successful HCCP access. In addition, the metal content of organometallic complexes was tested using ICP-MS and the results are listed in Table 3. The significant reduction of metal content after phosphonitrile modification laterally confirms that HCCP has been successfully incorporated into organometallic complexes, which confirmed that Zn-PDH was successfully synthesized according to the designed structure.

Comment 5: It is known that the introduction of dispersed fillers to an amount of more than 1 wt.% often leads to a decrease in the physical and mechanical characteristics of polymer composites. it would be nice to show how the filler you synthesized affects the strength of epoxy composites.

Response: Thanks for your precious suggestion. I couldn't agree with you more. The high-temperature curing epoxy thermoset materials were often required to possess certain mechanical properties. Generally, the compatibility of metallic compounds with organic polymers was not satisfactory, so the mechanical properties of materials would be greatly deteriorated after adding them. It is necessary to study the mechanical properties of the material. A similar compound organocobalt coordination complex (Co-H4APD) was previously prepared in our laboratory and investigated the effect of Co-H4APD on the mechanical properties of epoxy composites using Charpy impact test. (Journal of materials research and technology-JMR&T 2022, 21, 4921-4939; https://doi.org/10.1016/j.jmrt.2022.11.099). Indeed, the fracture impact strength value of the composite with the highly dispersed Co-H4APD had no significant change, but that of the sample with poorly dispersed CoO + H4APD drastically decreased by about 50%. This was mainly because the partial agglomerates of cobalt oxide in EP/CoO + H4APD could lead to crack initiation and propagation, consequently, produced the reduce strength of composites. Comparatively, the highly dispersed EP/Co-H4APD system was more efficient in transferring applied load than the poorly dispersed one, thus improving the impact resistance of them. Similarly, as the coordination reaction of Zinc ions and the existence of -NH functional groups, improving the dispersion of Zn-PDH and the interfacial interaction between flame retardants and matrix in the composites, the critical stress necessary to cause the flame retardant/matrix interfacial failure increased, thus lead to more fracture energy dissipation to resist the propagation of cracks during deformation, thus increasing the fracture toughness. Theoretically we presume that the Zn-PDH synthesized in our paper would have the same effect on the mechanical properties of the material as Co-H4APD. Therefore, its mechanical properties were not investigated in this paper.

Round 2

Reviewer 2 Report

The authors considered most of the comments or adequately responded to the remarks contained in the review; therefore, the work may be approved for publication.